# Gain-Guiding Anisotropic Polarized Amplified Spontaneous Emissions from C-Plane ZnO/ZnMgO Multiple Quantum Wells

**DOI:** 10.3390/ma15196668

**Published:** 2022-09-26

**Authors:** Ja-Hon Lin, Gung-Rong Chen, Sheng-Jie Li, Yu-Feng Song, Wei-Rein Liu

**Affiliations:** 1Department of Electro-Optical Engineering, National Taipei University of Technology, Taipei 10608, Taiwan; 2Intelligent Internet of Things and Intelligent Manufacturing Center, College of Electronics and Information Engineering, Shenzhen University, Shenzhen 518060, China; 3National Synchrotron Radiation Research Center, Hsinchu 30076, Taiwan

**Keywords:** stimulated emission, multiple-quantum well, exciton–exciton scattering, waveguide effect

## Abstract

A microcavity laser with linear polarization finds practical applications in metrology and biomedical imaging. Through a pulsed light excitation, the polarization characteristics of amplified spontaneous emissions (ASEs) from ten-period ZnO/Zn${}_{0.8}$Mg${}_{0.2}$O multiple quantum wells (MQWs) on a C-Plane sapphire substrate were investigated at room temperature. Unlike unpolarized spontaneous emissions, with 35 meV of energy differences between the C and AB bands, the ASE of MQWs revealed transverse-electric (TE) polarization under the edge emission configuration. The excited ASE from the surface normal of the polar ZnO/Zn${}_{0.8}$Mg${}_{0.2}$O MQWs with hexagonal symmetry revealed linear polarization under the pump of the stripe line through the focusing by using a cylindrical lens. The polarization direction of ASE is independent of the pump polarization but always perpendicular to the pump stripe, even if the cylindrical lens is rotated 90 degrees because of the gain-guiding effect.

## 1. Introduction

Zinc oxide (ZnO), a II-VI compound semiconductor with a hexagonal wurtzite structure, reference [1] has attracted considerable attention in the past several decades due to its promising applications in ultraviolet (UV) optoelectronic devices. ZnO has become one of the most promising materials in solar cells, light-emitting diodes (LEDs), laser diodes (LDs), etc., due to its direct wide bandgap of about 3.37 eV and large exciton binding energy of about 60 meV (RT). ZnO/ZnMgO multiple quantum wells (MQWs), in contrast to the intrinsic ZnO, possess superior advantages, such as tunable band-gaps, large exciton binding energy [2], and great improvements in radiative efficiency [3]. Thus, ZnO heterostructures [4,5] or ZnO-based quantum wells (QWs) [6] have been well-designed or actively developed to produce practical optoelectronic devices, such as highly efficient LEDs or LDs in the blue/UV spectrum. Scientists are also interested in carrier dynamics [7], acoustic phonon generation [8], and room temperature amplification spontaneous emission (RT-ASE) [2,9,10,11] from ZnO/ZnMgO MQWs. In previous reports, low-threshold RT-ASE from ZnO bulk, thin film [12,13], and MQWs [9] has been produced based on the exciton–exciton scattering (ex–ex scattering) instead of the electron-hole plasma recombination. Furthermore, the excitation threshold of ASE and the radiation lifetime of ZnO/ZnMgO MQWs have been investigated in relation to the good thickness [10] and Mg concentration [11].

On the other hand, the aluminum gallium nitride (AlGaN) MQW is a potential device used to produce a deep UV laser for application in high-density optical storage, water purification, and biomedical detection. In the analysis on the effect of a crystal-field split-off hole (CH) and heavy-hole (HH) band crossover on the gain characteristics of AlGaN QW with AlN barriers, a large TM-polarized material gain was achievable for high Al-content, causing feasible TM lasing at ∼220–230 nm [14]. Through metal–organic chemical vapor depositions, the transverse electric (TE)-polarization deep UV laser at 243 nm was produced from epitaxially grown from an AlGaN/AlN heterostructure on an Al-polar free-standing AlN (0001) substrate [15]. Although the TE polarized lasing characteristic has been demonstrated from 60 nm of thick ZnO nano-crystalline films (because of the optical waveguide effect [16]), the polarization feature of ASE from ZnO/ZnMgO MQWs has seldom been discussed. In addition, the ASE from the edge of the sample has been reported but its characteristics from the surface normal of the sample have rarely been investigated. Furthermore, the lattice-matched S${}_{c}$AlMgO${}_{4}$ substrate was adopted to improve the characteristics of ZnO-based superlattices for the investigation of ASE from ZnO/ZnMgO MQWs [9,11]. In this work, we studied the RT emission characteristics of high-quality ZnO/ZnMgO MQWs on a more cheap C-Plane sapphire substrate. In addition to the emission from the edge of the sample, we also studied the polarization characteristic of ASE from the surface normal direction of the polar ZnO/ZnMgO MQWs to inspect the underlying mechanism.

## 2. Results and Discussion

The ten-period ZnO/Zn${}_{0.8}$Mg${}_{0.2}$O MQWs, as illustrated in Figure 1a, were produced by the laser deposition [8]. In order to reduce crystal mismatching, the Zn${}_{0.95}$Mg${}_{0.05}$O buffer layer was grown on the C-Plane sapphire substrate first. Then, the ZnO (well) and Zn${}_{0.8}$Mg${}_{0.2}$O (barrier) were alternately deposited on top of the buffer layer. The high angle annular dark field scanning transmission electron microscopy (HAADF-STEM) image of ZnO/Zn${}_{0.8}$Mg${}_{0.2}$O MQWs, with high sensitivity to the variation in atomic species, is shown in the inset of Figure 1b. It illustrates the homogeneous epitaxial growth of our MQWs with good periodicity and the thicknesses of the well and barrier of around 2.5 and 13.1 nm, respectively. Owing to the larger lattice strain caused by the unrelaxed growth of the Zn${}_{0.95}$Mg${}_{0.05}$O buffer layer, the buckled or wrinkled structure is observed in the STEM image.

In Figure 1b, the high-resolution X-ray diffraction (XRD) shows the multiple periodic peaks from the diffraction of ZnO/Zn${}_{0.8}$Mg${}_{0.2}$O MQWs. The differences between two adjacent Bragg peaks, Δq = 0.0398 (1/A${}^{\circ }$), can be used to estimate the period (Λ) of MQWs via the formula: $\Lambda =\frac{2\pi }{\Delta q}$. The estimated period of MQWs Λ= 15.8 nm is close to the value Λ = 15.6 nm from the STEM image. The cryostat photoluminescence (PL) measurement of the MQWs was measured using a He-Cd laser as an excitation source with a central wavelength of 325 nm. The backscattering emission light was guided into a monochromator (iHR 550, Horiba, Inc., Kyoto, Japan) and then detected by a photo-multiplier (Horiba, Inc.). Figure 1c shows the absorbance (navy solid curve) and RT-PL (red solid curve) spectra of the ZnO/ZnMgO MQWs. The band edge of ZnO, $E_{g}$ = 3.484 eV, can be obtained from the shoulder of the absorbance spectrum. In addition, the emission peak of RT-PL illustrates that the free exciton (FX) energy ($E_{ex}$) of ZnO is around 3.394 eV. Thus, we derived the binding energy ($E_{b}$) of the ZnO/Zn${}_{0.8}$Mg${}_{0.2}$O MQWs of about 90 meV from the difference between the $E_{g}$ and $E_{ex}$.

The inset of Figure 1d shows the temperature-dependent PL from 15 to 300 K. It reveals two main emission peaks resulting from ZnO and ZnMgO, respectively, and subsequent peaks resulting from the longitudinal optical (LO) phonon replicas. The integrated intensities of free exciton emissions from ZnO versus reciprocal temperature (1/T) can be well fitted by the equation [1]:(1)$$I\left(T\right)=\frac{I_{0}}{1+a_{1}e^{\frac{-E_{a1}}{k_{B}T}}+a_{2}e^{\frac{-E_{a2}}{k_{B}T}}},$$
where $I_{0}$ = 5.04 is the integrated intensity at 0 K, $k_{B}$ is Boltzmann’s constant, $E_{a1}$= 90.0 meV and $E_{a2}$= 12.3 meV are the exciton binding energy and the trapping energy of the exciton bound to the neutral donor or localization potential, and $a_{1}$ and $a_{2}$ are the corresponding constants. In contrast to the bulk with $E_{b}$ = 60 meV, the derived exciton binding energy of the C-Plane ZnO/Zn${}_{0.8}$Mg${}_{0.2}$O MQWs was around 90 meV. It was attributed to the quantum confinement (QC) effect when the well width was smaller than 3 nm [17].

Figure 2a illustrates the experimental setup to obtain the polarization characteristic of the edge emission from the MQWs. A frequency-quadrupled Q-switched Nd:YAG laser with a central wavelength of 266 nm was used as a pump source, which revealed a 7 ns pulse width and 10 Hz repetition rate. A half-waveplate (λ/2 WP) in combination with a polarization beam splitter (PBS) was used to control the excited pulse energy onto the MQWs. A cylindrical lens with a focal length of 7.5 cm was used to focus the pump beam onto the MQWs with a long stripe (5.57 × 0.11 mm). The edge emission of the MQWs was collected by a lens and launched into a fiber. A linear polarizer (LP) was put in front of the fiber. The emission spectrum with perpendicular polarization (*E* ⊥ c), i.e, TE mode (θ = $0^{\circ }$ and $180^{\circ }$) and parallel polarization direction (E ‖ c), i.e., TM mode (θ = $90^{\circ }$, and $270^{\circ }$), relative to the c-axis of ZnO, could be deduced from this measurement. The emission light of the MQWs was dispersed by a monochromator (iHR 320, Horiba, Inc.) and then detected by an electrically cooled charge-coupled device (CCD) (Syncerity, Horiba, Inc.). From the selection rule, the transition from the conduction band to the valence band in ZnO bulk can be illustrated with three bands (A, B, and C) as shown in Figure 2b. In our measurement, the TE mode (*E* ⊥ c) corresponds to the A and B exciton transitions (FX-A,B) and the TM mode (*E* ‖ c) corresponds to the C exciton transition (FX-C).

The inset of Figure 2c shows the spontaneous emission (SPE) of the MQWs as the polarization varied from $0^{\circ }$ (*E* ⊥ c) to $90^{\circ }$ (*E* ‖ c) at a low pump intensity $I_{pump}$ of about 0.24 MW/cm${}^{2}$. The peak photon energy of SPE revealed a blue shift of about 35 meV from the TE mode (∼3.298 eV) to the TM mode (∼3.333 eV), which corresponded to the energy difference between the AB- and C-bands as shown in Figure 2b. At a high $I_{pump}$ of about 1.79 MW/cm${}^{2}$, the variation of ASE from the MQWs (as θ increased from $0^{\circ }$ to $90^{\circ }$) is shown in Figure 2c. The photon energy was almost fixed at around 3.287 eV, but the peak intensity decreased as ASE varied from the TE ($\theta =0^{\circ }$) to the TM ($\theta =90^{\circ }$) mode. When the pump intensity increased, the peak intensity of the ASE from MQWs (*E* ⊥ c) as a function of the $I_{pump}$ is shown in Figure 2d. After linear fitting, the threshold could be obtained by the intersection of two straight lines. It indicates that the ASE threshold of the TE mode (*E* ⊥ c) is about 0.5 MW/cm${}^{2}$. The logarithm peak intensity of ASE as a function of $I_{pump}$ (inset of Figure 2d) can be well-fitted by the equation [18]: $I_{out}=\alpha I_{pump}^{n}$. In contrast to SPE with n≃1, the larger exponent n≃2 demonstrates that the ASE could result from the ex–ex scattering. The full width at half maximum (FWHM) of ASE (*E* ⊥ c) from the edge of the MQW is also revealed by red squares, as shown in Figure 2d. It shows a clear decrease of FWHM around the thresholds and an obvious increase in the peak intensity of ASE. The FWHM of ASE above the threshold was almost a constant value of around 31 meV.

The polar plot of peak intensity and photon energy of SPE from the edge emission of the MQW are shown in Figure 3a,b, respectively. In Figure 3a, the peak intensity is almost the same under different polarization directions, which exhibits the unpolarized SPE from the edge emission of the MQWs. However, the variation of photon energy under different polarization states reveals a dumbbell shape in Figure 3b. Figure 3c,d illustrate the polar plot of peak intensity and photon energy of ASE from the edge emission of the MQWs. In contrast to the SPE in Figure 3a, a dumbbell shape of the peak intensity distribution in Figure 3c indicates that the maximum and minimum intensities of the ASE component are perpendicular (θ = $0^{\circ }$ and $180^{\circ }$) and parallel (θ = $90^{\circ }$ and $270^{\circ }$) to the c-axis of the MQWs, respectively. This is attributed to the larger gain for the TE mode transition between the conduction band and the A or B valence band in the ZnO/ZnMgO MQWs. The linear polarized ASE from the edge emission of the MQW with a degree of polarization (*P*) of about 0.95 was estimated by the formula [16]: $P=\frac{I_{TE}-I_{TM}}{I_{TE}+I_{TM}}$, where $I_{TE}$ is the perpendicular (TE mode) and $I_{TM}$ is the parallel (TM mode) intensity component. Figure 3d shows that the photon energy of the ASE peak from the edge of ZnO/ZnMgO MQWs is almost fixed at 3.287 eV.

Table 1 presents the reported ASE from ZnO thin film and ZnO/ZnMgO MQWs. The emission from the surface normal of the ZnO/ZnMgO MQWs was rarely investigated. In order to investigate the ASE of the MQWs, we further performed polarization-dependent emissions from the surface normal of the sample, as shown in Figure 4a. One would expect the isotropic distributions for the normal ASE and SPE spectra because of the hexagonal symmetry of ZnO. The TM mode pump polarization (solid arrow line in Figure 4a) was reflected by the dichroic mirror (DM) and then focused on the cylindrical lens to form a stripe line (along the x-axis) onto the MQWs. The emission spectra from the MQWs were collimated by a cylindrical lens (once again) and then passed through the DM. A LP was put in front of the fiber tip to distinguish the electric field perpendicular ($\theta \;=\;0^{\circ }$ and $180^{\circ }$) or parallel ( $\theta \;=\;90^{\circ }$ and 270${}^{\circ }$) to the stripe line. Here, the angle θ is the included angle between the polarization of the LP and the y-direction (perpendicular to the direction of the long stripe). Figure 4b shows the ASE spectra of the MQWs under the high pump intensity $I_{pump}$ = 7.62 MW/cm${}^{2}$. In addition, the SPE spectra of MQWs with low pump energy densities $I_{pump}$ = 0.71 MW/cm${}^{2}$ are also shown in the inset of Figure 4b. The emission spectrum can be decomposed into two Gaussian functions to show two peaks at 3.41 and 3.55 eV that are close to the FX-C emission peak of the ZnO and ZnMgO from RT-PL in Figure 1 c. Similarly, the emission spectrum in Figure 4b can be decomposed into three emission peaks resulting from the ASE at 3.28 eV (red curve), SPE of ZnO at 3.32 eV (blue curve), and SPE of ZnMgO at 3.53 eV (green curve). The red shift of the ASE peak relative to the SPE could be attributed to the band gap renormalization or self-absorption [19,20].

Due to the hexagonal symmetry of the C-Plane ZnO/ZnMgO with a honeycomb structure, an unpolarized normal SPE from the MQWs is demonstrated, as expected, and shows a fixed peak photon energy at about 3.408 eV under different polarization states (not shown here). Nevertheless, a linear polarized ASE with *P*≃ 0.85 was revealed from the polar plot of the peak intensity in Figure 4c, which shows a dumbbell-shaped intensity distribution under $I_{pum}$ = 7.62 MW/cm${}^{2}$. The maximum output intensity of ASE is perpendicular to the long stripe of the pump beam (θ = $0^{\circ }$ and $180^{\circ }$). Similar to the case of edge emission (Figure 3d), the peak photon energy of ASE is still fixed at around 3.284 eV and does not vary with the rotation of the LP as shown in the polar plot of Figure 4d. This finding implies that the detected polarized ASE is possibly due to the scattering of the amplified light along the pump stripe.

In order to verify this assumption, we first changed the incident pump beam from the TM to the TE mode (dashed arrow in Figure 4a). In Figure 5a, the output ASE still preserves the TE mode, with a degree of polarization of about 0.89, so that the influence of the pump polarization can be excluded. Then, we changed the pump scheme by keeping the pump polarization in TM mode but rotated the cylindrical lens to make the direction of the focused stripe line along the y-direction onto the MQWs (Figure 5d). The polar plot of the intensity distribution from the MQWs in Figure 5b illustrated that the maximum peak intensity occurred at θ = $90^{\circ }$ and $270^{\circ }$. This indicates that the ASE was changed to the TM mode with *P* ≃ −0.89. The pump stripe produced an optical waveguide effect on the propagation ASE with polarization perpendicular to the pump stripe (the waveguide mode). The recorded normal ASE is a result of scattered photons from the sample surface and measured in our system. Therefore, we observed that the polarization of ASE was always perpendicular to the pump stripe, even if the cylindrical lens was rotated 90 degrees as shown in Figure 5c,d. The pump stripe line on ZnO/Zn${}_{0.8}$Mg${}_{0.2}$O broke the hexagonal symmetry of ZnO to produce the anisotropic polarized ASE from the surface normal.

## 3. Conclusions

In summary, we demonstrated the linear polarization characteristics of amplified spontaneous emission (ASE) from optical-pumped ZnO/Zn${}_{0.8}$Mg${}_{0.2}$O MQWs on C-Plane sapphire at room temperature. The MQWs was produced through pulse laser deposition to alternatively grow the 10 periods of ZnO (well) and ZnMgO (barrier) layers, respectively. Through the configuration of the edge emission from the MQWs, the unpolarized spontaneous emission was obtained and showed a shift in photon energy from the AB-band to the C-band at about 35 meV. Nevertheless, transverse-electric polarized ASE at 3.28 eV, perpendicular to the c-axis of ZnO, was demonstrated from the edge emission of the MQWs under high-excited pump energy. Beyond our expectations, linear polarized ASE from the surface normal of the C-Plane MQWs was excited using a cylindrical lens to ’focus’ the pump stripe line on the sample. The polarized direction of ASE was independent of the pump polarization but normal to the long pump stripe, which could be attributed to the gain guiding effect.

## Figures and Tables

**Figure 1 materials-15-06668-f001:**
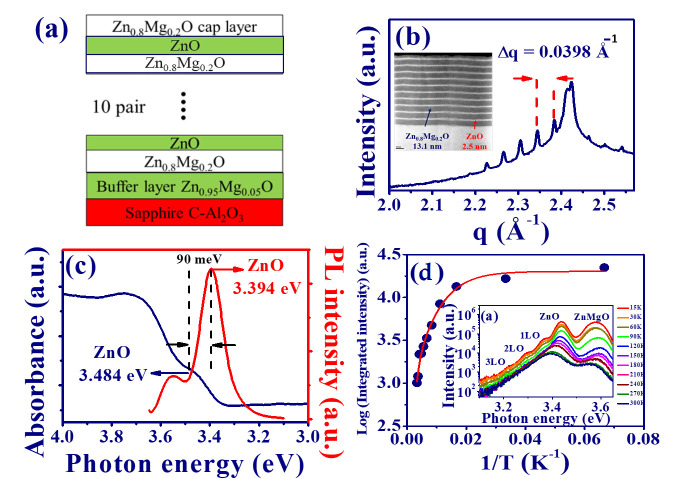
(**a**) Schematic illustration of the structure of the ZnO/Zn${}_{0.8}$Mg${}_{0.2}$O MQWs, (**b**) X-ray diffraction image (Inset: HAADF-STEM image of the MQWs), (**c**) absorbance (navy solid line), RT-PL spectrum (red solid line), and (**d**) integrated PL intensity as functions of (1/*T*) (Inset:temperature dependent PL spectrum) of the ZnO/ZnMgO MQWs.

**Figure 2 materials-15-06668-f002:**
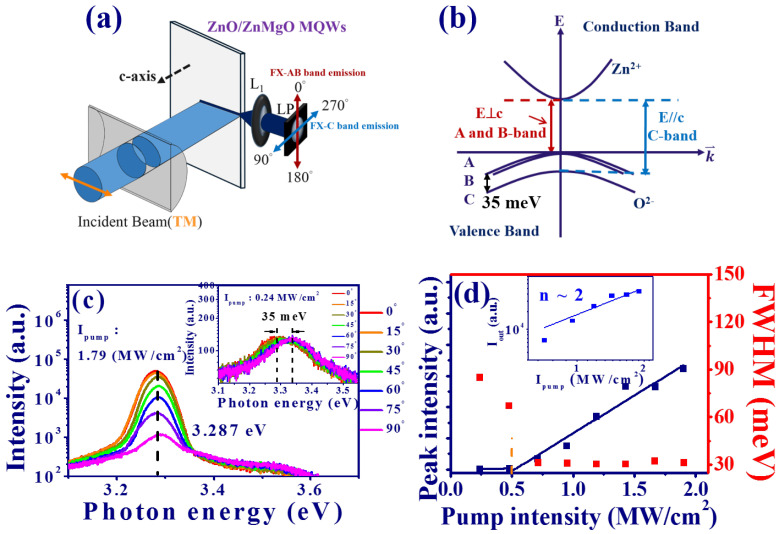
(**a**) Schematic illustration of the edge emission from the MQWs through the pump of a Q-switched laser, (**b**) band diagram of ZnO. (**c**) Polarization-dependent ASE and SPE spectra (inset) as θ increases from $0^{\circ }$ to $90^{\circ }$, and (**d**) peak intensity and FWHM of the emission spectrum from the MQWs versus pump intensity using the TM mode pump polarization. (Inset: logarithm intensity of ASE as a function of pump intensity. )

**Figure 3 materials-15-06668-f003:**
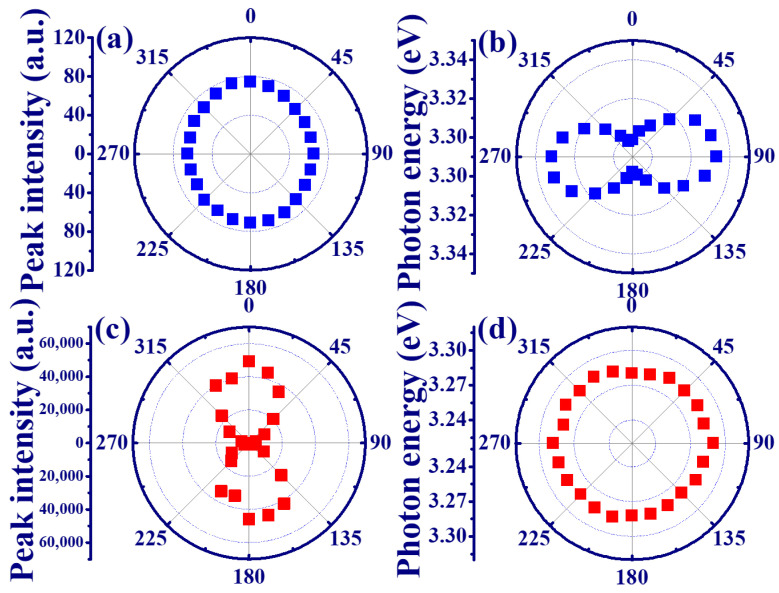
Polar plots of (**a**) peak intensity and (**b**) photon energy of the SPE as well as the (**c**) peak intensity and (**d**) photon energy of the ASE from the edge emission of the MQWs.

**Figure 4 materials-15-06668-f004:**
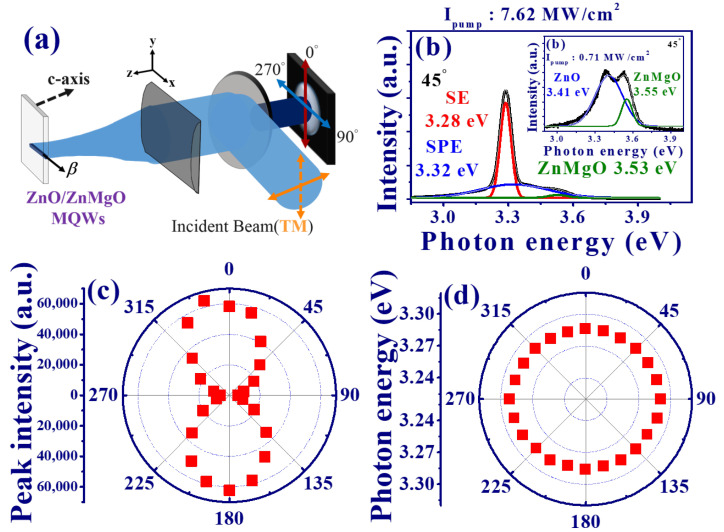
(**a**) Schematic illustration of the normal emission from MQWs through the pump of a Q-switched laser. (**b**) ASE spectrum from the surface normal of MQWs at θ = $45^{\circ }$. (Inset: SPE spectrum), (**c**) peak intensity and, (**d**) photon energy of ASE from the surface normal of MQWs (TM mode pump polarization).

**Figure 5 materials-15-06668-f005:**
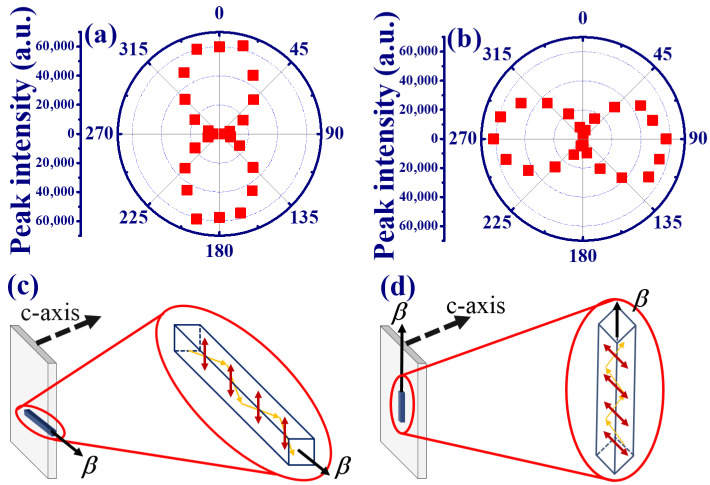
Polar plots of peak intensities from the surface normal of the MQWs by the (**a**) TE mode pump polarization, and (**b**) the TM mode pump polarization after the rotation of a cylindrical lens by 90 degrees. Illustration of the light propagation and direction of the electric field of SE by the (**c**) horizontal and (**d**) vertical pump stripe line on the MQWs.

**Table 1 materials-15-06668-t001:** Investigation of ASE from ZnO and ZnO/ZnMgO MQWs.

Sample	Substrate	Preparation	RT ASE Peak	Emission	Ref.
		Method	(eV)		
ZnO/Zn${}_{0.88}$Mg${}_{0.12}$O	S*c*AlMgO${}_{4}$	L-MBE		Side	[9]
MQWs	(0001)				
ZnO/Zn${}_{0.82}$Mg${}_{0.18}$O	Sapphire	PLD	3.14		[10]
MQWs	(0001)				
ZnO/Zn${}_{0.88}$Mg${}_{0.12}$O	S*c*AlMgO${}_{4}$	L-MBE	3.24	Side	[11]
MQWs					
ZnO	Sapphire	L-MBE	3.20	Side	[12]
thin film	(0001)				
ZnO	Sapphire	PE-MBE	3.03	Side	[13]
thin film	(0001)				
ZnO/Zn${}_{0.8}$Mg${}_{0.2}$O	Sapphire	PLD	3.28	Side/Normal	Our
MQWs	(0001)				work

## Data Availability

Not applicable.

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
