# Peer review of "Gain-Guiding Anisotropic Polarized Amplified Spontaneous Emissions from C-Plane ZnO/ZnMgO Multiple Quantum Wells"

_materials, 2022, doi:10.3390/ma15196668_

Round 1

Reviewer 1 Report

The Authors Lin et al presented a manuscript with the title of “Gain-guiding Anisotropic Polarized Amplification Spontaneous Emissions from C-plane ZnO/ZnMgO Multiple Quantum Wells”. The authors studied the polarization characteristics of emission from ZnO/ZnMgO MQWs on c-plane sapphire substrate. 

The manuscript is very well written. The experimental work has been well planned. This work may be interested to the researchers/readers in the related community. This draft can be accepted for publication if below comments can be clarified.

My comments are list as below.

1. Fig. 2: what is the dimension of the sample? If the incident beam (5.57 mm × 0.11 mm) is smaller than the sample (width), at least part of the excited emission will be absorbed by the un-pumped regions. In addition, is there any waveguide along the direction of the edge light emission? Otherwise, the excited light will come out in all directions. With this setup, part of the light will be missed during measurement, therefore, the results in Fig. 2d, and inset of Fig. 2 d is questionable.

2. Fig. 2(a): the incident beam (TM.), did the authors specially choose TM polarization of the incident beam? However, for the excitation light, TE and TM has no difference as long as the light is normal to the wafer surface (and excited light comes out from edge side). 

3. Fig. 2 (b) (starting from line 88) explanation is not accurate for describing QW transition. With QW used in this work, the light emission should be electron level transit to hole level in the QW.

4. Fig. 4 (a), the authors changed the TM and TE of incident light, however, all electrical polarization of the incident light lies in the wafer plane. In theory, TM or TE of incident light has nothing to do with emission from the studied samples. Can the authors explain on this?

End of review.

Author Response

  1. 2: what is the dimension of the sample? If the incident beam (5.57 mm × 0.11 mm) is smaller than the sample (width), at least part of the excited emission will be absorbed by the un-pumped regions. In addition, is there any waveguide along the direction of the edge light emission? Otherwise, the excited light will come out in all directions. With this setup, part of the light will be missed during measurement, therefore, the results in Fig. 2d, and inset of Fig. 2 d is questionable.

Reply:

The dimension of the sample is around 1.2 cm x 0.7 cm as shown in following plot.  In addition, the waveguide effect will be produced under the excitation of pump pulses.  As mentioned from the reviewer, not all the emission light from the sample will be measured in our detection system.  Nevertheless, Fig. 2(d) can be used to demonstrate the lasing behavior and shows the threshold of ZnO/ZnMgO multiple quantum well (MQW) under the excitation of Q-switched laser.  The inset of Fig. 2(d) is used to obtain the exponent around 2 and demonstrate the ex-ex scattering effect.  Actually, the unit in Y-axis is arbitrary in Fig. 2.  Thanks for the comment. 

  1. 2(a): the incident beam (TM.), did the authors specially choose TM polarization of the incident beam? However, for the excitation light, TE and TM has no difference as long as the light is normal to the wafer surface (and excited light comes out from edge side).

Reply: No, the incident pump beam with TM mode was not specially choose for the side emission of Fig. 2(a). The reviewer is absolutely right.  The TE and TM pump polarization has no influence for the side emission of ZnO/Zn0.8Mg0.2O multiple quantum wells (MQWs) grown on a c-plane sapphire substrate.

  1. 2 (b) (starting from line 88) explanation is not accurate for describing QW transition. With QW used in this work, the light emission should be electron level transit to hole level in the QW.

Reply:

Actually, the band diagram in Fig. 2(b) cannot be used to describe the transition in QW.  Owing to more complicated mechanism for the light emission in QW, the band-diagram of ZnO bulk in Fig. 2(b) has been used instead of the QWs for describing the anisotropic property and degenerate valence band.  We have modified the sentence as “From the selection rule, the transition from the conduction band to the valence band in ZnO bulk can be illustrated with three bands (A, B and C) as shown in Fig. 2(b).”  Thanks for the reminding.

  1. Fig. 4 (a), the authors changed the TM and TE of incident light, however, all electrical polarization of the incident light lies in the wafer plane. In theory, TM or TE of incident light has nothing to do with emission from the studied samples. Can the authors explain on this?

Reply:

The reviewer is absolutely right.  For the ZnO/ZnMgO MQW on c-plane sapphire, the TE and TM pump polarization has nothing to do with the result of our measurement.  In order to investigate the underlying mechanism of linear polarized ASE generation in Fig. 4(c) through the stripe pump line on sample, we change the pump polarization from TM mode to TE mode in this work.  As expected, the polarization direction of ASE in Fig. 5(a) is similar to the result in Fig. 4(c).  Thanks for the comment.

Reviewer 2 Report

To comprehend the ASE performance, the authors created the hybrid structure from C-plane ZnO/ZnMgO Multiple Quantum Wells. The results in the manuscript were adequately conveyed as per the title. Some missing motivation and several typos in this study, led me to conclude that a minor revision is needed before it is to be published in Materials journal.

1.     The introduction of AlGaN in the context of this manuscript is not well described.

2.     Need to elaborate more on the motivation of this work in the introduction.

3.     Why particularly 10 pairs were chosen; any explanations for this?

4.     There are several typos present. Please be careful while resubmitting it. For example, a) title – ASE stands for “Amplified Spontaneous Emissions” and not Amplification Spontaneous Emissions – please correct

b) Abstract – should it be “Microcavity”? not Microcaity.

Please check thoroughly.

5.     A Table with the comparison between similar systems/compositions about ASE would certainly help.

Author Response

  1. The introduction of AlGaN in the context of this manuscript is not well described.

Reply:

We have added more detail description of AlGaN on page 1 in our manuscript as following “On the other hand, the aluminum gallium nitride (AlGaN) MQW is a potential device to produce a deep UV laser for the application in high-density optical storage, water purification, and bio-medical detection.  In analysis of the effect of crystal-field split-off hole (CH) and heavy-hole (HH) bands crossover on the gain characteristics of AlGaN QW with AlN barriers, large TM-polarized material gain is achievable for high Al-content to cause feasible TM lasing at ~220–230 nm [14].  Through metalorganic chemical vapor deposition, the transverse electric (TE)-polarization deep UV laser at 243 nm was produced from epitaxially grown of AlGaN/AlN hetrostructure on an Al-polar free-standing AlN (0001) substrate [15].” Thanks for the reminding.

  1. Need to elaborate more on the motivation of this work in the introduction.

Reply:

The motivation has been modified on page 2 in our resubmitted manuscript as following “Although the TE polarized lasing characteristic has been demonstrated from the 60 nm thick ZnO nano-crystalline films because of the optical waveguide effect [16], the polarization feature of ASE from ZnO/ZnMgO MQWs has seldom been discussed.  In addition, the ASE from edge of sample was reported but rare discussion of its characteristic from surface normal of the sample. Furthermore, the lattice lattice-matched ScAlMgO4 substrate was adopted to improve the characteristics of ZnO based superlattices for the investigation of ASE from ZnO/ZnMgO MQWs [9, 11]. In this work, we studied the RT emission characteristics of high quality ZnO/ZnMgO MQWs on a more-cheap c-plane sapphire substrate. In addition to the emission from edge of sample, we also studied the polarization characteristic of ASE coming from the surface normal direction of the polar ZnO/ZnMgO MQWs to inspect the underlying mechanism.

  1. Why particularly 10 pairs were chosen; any explanations for this?

Reply:

Theoretically, the MQWs reveal higher confinement effect to avoid tunnel effect, and higher photoluminescence efficiency as the grown period of ZnO and ZnMgO layer above 10.  In addition, the 10 pairs ZnO/ZnMgO MQWs have been produced previously to investigate the exciton binding energy enhancement and the characteristic of ASE in early report.  In order to compare the carrier and exciton dynamics with previous results, we adopted the 10 pair ZnO/ZnMgO for this study.

  1. There are several typos present. Please be careful while resubmitting it. For example, a) title – ASE stands for “Amplified Spontaneous Emissions” and not Amplification Spontaneous Emissions – please correct
  2. b) Abstract – should it be “Microcavity”? not Microcaity.

Please check thoroughly.

Reply:

We have modified these typos in our revised manuscript. Thank you for kind reminding.

  1. A Table with the comparison between similar systems/compositions about ASE would certainly help.

Reply:

A table 1 has been added in our revised manuscript to compare the ASE generated from ZnO thin film and ZnO/ZnMgO MQWs.  Thanks for the suggestion.
